# AUTOMATIC NEURAL SPATIAL INTEGRATION

## ABSTRACT

Spatial integration is essential for a number of scientific computing applications, such as solving Partial Differential Equations. Numerically computing a spatial integration is usually done via Monte Carlo methods, which produce accurate and unbiased results. However, they can be slow since it require evaluating the integration many times to achieve accurate low-variance results. Recently, researchers have proposed to use neural networks to approximate integration results. While networks are very fast to infer in test-time, they can only approximate the integration results and thus produce biased estimations. In this paper, we propose to combine these two complementary classes of methods to create a fast and unbiased estimator. The key idea is instead of relying on the neural network's approximate output directly, we use the network as a control variate for the Monte Carlo estimator. We propose a principal way to construct such estimators and derive a training object that can minimize its variance. We also provide preliminary results showing our proposed estimator can both reduce the variance of Monte Carlo PDE solvers and produce unbiased results in solving Laplace and Poisson equations.

## 1 INTRODUCTION

In this paper, we are interested in numerically estimating a family of spatial integrations:

$$F(\mathbf{z}) = \int_{\Omega(\mathbf{z})} f(\mathbf{p}, \mathbf{z}) d\mathbf{p}, \tag{1}$$

where $\Omega \subset \mathbb{R}^d$ denotes a domain to integrate over, $f : \mathbb{R}^d \times \mathbb{R}^h \to \mathbb{R}$ is the integrands, and $\mathbf{z}$ is a vector parameterizing of the family of integral. We assume the domain $\Omega(\mathbf{z})$ to be structured and parameterizable (e.g. 3D spheres with different centers). The goal is to numerically estimate $F(\mathbf{z})$ accurately and efficiently via samples of $(\mathbf{p}, f(\mathbf{p}))$'s, for all $\mathbf{z}$ of interests.

Computing such spatial integration is important for many applications in scientific computing and computer graphics. For example, producing physics-based rendering from 3D shapes requires integrating light sources from different incoming directions (Veach, 1998). Solving partial differential equations using integral equations also needs to integrate over various spherical domains (Sawhney & Crane, 2020). In these applications, every query can result in thousands of spatial integrations over different domains, and users usually need thousands of queries to obtain meaningful information. As a result, being able to estimate spatial integrals efficiently is very important.

A common approach to estimate these integrals is via Monte Carlo methods (Veach, 1998; Spanier & Gelbard, 2008; Sawhney & Crane, 2020). Monte Carlo methods first rewrites the integral into an expectation, which can then be estimated via sampling. Specifically, for a given $\mathbf{z}$, we have:

$$\int_{\Omega(\mathbf{z})} f(\mathbf{p}, \mathbf{z}) d\mathbf{p} = \mathbb{E}_{\mathbf{P} \sim P_{\Omega(\mathbf{z})}} \left[ \frac{f(\mathbf{p}, \mathbf{z})}{P_\Omega(\mathbf{p})} \right] \approx \frac{1}{N} \sum_{i=1}^{N} \frac{f(\mathbf{p}_i, \mathbf{z})}{P_{\Omega(\mathbf{z})}(\mathbf{p}_i)}, \tag{2}$$

where $P_\Omega$ is the sampling distribution defined on the domain $\Omega(\mathbf{z})$ and $\mathbf{p}_i \sim P_{\Omega(\mathbf{z})}$ are independent samples from the distribution. While Monte Carlo methods are unbiased, the variance of the estimator decays at the rate of $O(\frac{1}{N})$. As a result, obtaining accurate outcomes from Monte Carlo requires a lot of independent samples of $f$ and $P_\Omega$. This makes the method slow when evaluating $f$ or sampling from $P_\Omega$ is expensive.

An emerging alternative is using deep neural networks to approximate the output of these integrals (Lindell et al., 2021; Maître & Santos-Mateos, 2023). These methods optimize a neural network $G_\theta$

to approximate a family of integrals by matching $G_\theta$'s derivative with the integrands. For example, AutoInt (Lindell et al., 2021) considers all possible line integrals in the following form: $F(a, b) = \int_a^b f(x)dx$ for $L \leq a < b \leq U$, where $L$ and $U$ defines the domain of interests. AutoInt defines a network $G_\theta$ and use $G_\theta(a) - G_\theta(b)$ to approximate $F(a, b)$. The optimal $\theta$ is obtained by minimizes the loss $\mathbb{E}_{x \in [L,U]} \left[ \|G'_\theta(x) - f(x)\|^2 \right]$. Once trained, AutoInt can approximate a family of integrals very efficiently with only a few network forward passes. However, finding the optimal parameters that make $G'_\theta(x) = f(x)$ for all possible $x$ is nearly impossible due to limitations in computation or network capacity. Thus, once the network is trained, it can produce potentially biased solutions. It remains unclear whether such bias can be rectified as more computational resources and data become available.

Given the complementary properties of Monte Carlo and neural methods, the following question arises: *Can we develop a method that is both quick in inference and also assures unbiased results, given that sufficient computing resources are available?* In this paper, we hypothesize that this can be achieved by applying automatic neural integration for control variates. The key idea is that, instead of using the network's output as the final result, we also account for its error. As long as we can construct two computational graphs $G_\theta$ and $\partial G_\theta$ such that $G_\theta(\Omega) = \int_\Omega \partial G_\theta(\mathbf{p})d\mathbf{p}$, the following identity holds:

$$\int_\Omega f(\mathbf{p})d\mathbf{p} = G_\theta(\Omega) + \int_\Omega f(\mathbf{p}) - \partial G_\theta(\mathbf{p})d\mathbf{p} = G_\theta(\Omega) + \mathbb{E}_{P_\Omega} \left[ \frac{f(\mathbf{p}) - \partial G_\theta(\mathbf{p})}{P_\Omega(\mathbf{p})} \right], \quad (3)$$

where $P_\Omega$ is a probability distribution on domain $\Omega$, from which we can sample and compute density. The latter part of the integration can be estimated using the Monte Carlo method. The key insight is that we can derive a training objective for $\theta$ to minimize the variance of this new Monte Carlo estimator. The resulting estimator will require fewer samples to achieve the same accuracy while remaining unbiased, as the equality holds as long as $G_\theta(\Omega) = \int_\Omega \partial G_\theta(\mathbf{p})d\mathbf{p}$.

In this paper, we provide a proof of concept that this idea can indeed create an unbiased and lower variance estimator for spatial integrals. We first derive a principal way to extend the neural integration methods to spatial integration. We then use control variates techniques to construct an unbiased estimator using these neural networks. We also derive the training objective that can minimize the variance of this estimator. We test the effectiveness of our methods in Monte Carlo PDE solvers (Sawhney & Crane, 2020; Sawhney et al., 2022). Preliminary results prove that our proposed method is unbiased by construction and experiences fewer variances in these applications.

## 2    RELATED WORK

Our paper mainly draws inspiration from two lines of work: Monte Carlo and neural network integration methods. We will focus on reviewing the most relevant papers in those two lines of work and refer readers to Solomon (2015) for other numerical integration methods.

**Monte Carlo integration.**    Monte Carlo integration is very general and it has been applied to a large number of applications including physics-based rendering (Veach, 1998), solving partial differential equations (Sawhney et al., 2022; Sawhney & Crane, 2020), and various physics simulations such neutron transports (Spanier & Gelbard, 2008; Lewis & Miller, 1984) and fluid simulation (Rioux-Lavoie et al., 2022). Despite its versatility and unbiased nature, a significant drawback of Monte Carlo estimators is their high variance. To address this, numerous efforts aim to reduce variance through methods such as caching (Miller et al., 2023; Müller et al., 2021), importance sampling (Müller et al., 2019; Veach & Guibas, 1995), and control variates . Among these methods, control variates are particularly relevant to our work, achieving lower variance by computing the difference between the original random variable and another random variable with known integral values. Prior works have applied control variates in many applications including option pricing (Ech-Chafiq et al., 2021), variational inference (Geffner & Domke, 2018; Wan et al., 2019), and Poisson image reconstruction (Rousselle et al., 2016). To establish a control variate, we need to find a function that both has a known analytical integration and approximates the integrand function well. Most prior works usually construct the control variate heuristically (Lafortune & Willems, 1994; Clarberg & Akenine-Möller, 2008; Kutz et al., 2017). Such an approach can be difficult to generalize to complex integrands. One way to circumvent such an issue is to make the control variates learnable and

optimize the control variate function using samples from the integrand (Vévoda et al., 2018). For example, Salaün et al. (2022) proposed to use a polynomial-based estimator as control variate as the integration of the polynomial basis is easy to obtain. Recently, Müller et al. (2020) proposed to use normalizing flow as the control variate function since normalizing flows are guaranteed to integrate into one. Our method extends these works by expanding the choice of estimator family to a broader class of neural network architecture. In addition, we focus on applying this technique to solving PDEs using Walk-on-sphere methods Sawhney & Crane (2020); Sawhney et al. (2022; 2023).

**Neural Network Integration Methods.** Deep learning has emerged as a dominant optimization tool for many applications, particularly for numerical integration estimation. A prevalent strategy involves crafting specialized neural network architectures with analytical integration capabilities, similar in spirit to the Risch or Risch-Norman algorithm (Risch, 1969; Norman & Moore, 1977). For example, normalizing flows (Tabak & Turner, 2013; Dinh et al., 2016; Chen et al., 2018; Dinh et al., 2014) is a family of network architectures that models an invertible mapping, which allows them to model probability distribution by integrating into one. Other examples include Petrosyan et al. (2020) and Subr (2021), which designed network architectures that can be integrated analytically. These approaches usually result in a limited choice of network architectures, which might limit the expressivity of the approximator. An alternative approach is to create computational graphs that can be integrated into a known network by taking derivatives. For example, Nsampi et al. (2023) leverages repeated differentiation to compute convolutions of a signal represented by a network. In this work, we follow the paradigm proposed by AutoInt (Lindell et al., 2021), where we construct the integrand by taking derivatives of the network approximating the integration result. This approach can allow a more flexible choice of network architectures, and it has been successfully applied to other applications such as learning continuous time point processes Zhou & Yu (2023). Unlike the Monte Carlo method, a potential drawback to the AutoInt method is that it can create biased estimations. In this work, we propose to combine the AutoInt method with neural control variate to create an unbiased estimator.

## 3 BACKGROUND

**Problem set-up.** In this paper, the spatial integration we are interested in takes the following form:

$$F(\mathbf{z}) = \int_{\Omega(\mathbf{z})} f(\mathbf{p}, \mathbf{z}) d\mathbf{p}, \tag{4}$$

where $\mathbf{z} \in \mathbb{R}^h$ is a latent vector parameterizing a family of integration domains, $\Omega(\mathbf{z}) \subset \mathbb{R}^d$ defines a region where we would like to integrate and $f : \mathbb{R}^d \times \mathbb{R}^h \to \mathbb{R}$ is a function that can be queried within the domain $\Omega(\mathbf{z})$, and $d\mathbf{p}$ is the differential element. We assume there exists a parameterization of the region $\Omega$, which is a differentiable and invertible function that maps a region of $\mathbb{R}^d$ to points inside the domain $\Omega$: $\forall \mathbf{z}, \Phi(\mathbf{z}) : [l_1, u_1] \times \cdots \times [l_d, u_d] \leftrightarrow \Omega$. Intuitively, the mapping $\Phi(\mathbf{z})$ describes how to map the shape of a rectangular space into the integration domain of interest. This allows us to transform the integration into a more regular domain.

Different applications call for different form of domain $\Omega$'s. In physics-based rendering, one usually needs to integrate over all solid angles on a hemisphere (Veach, 1998) In this case, $\Omega(\mathbf{z})$ can be defined as spheres centered at a surface intersection point $\mathbf{z} \in \mathbb{R}^3$: $\{\mathbf{x} | \mathbf{x} \in \mathbb{R}^3, \|\mathbf{x} - \mathbf{z}\| = 1\}$. The mapping $\Phi$ can be defined as: $\Phi([\theta, \phi]^T) = [\cos(\theta)\sin(\phi), \sin(\theta)\cos(\phi), \cos(\phi)]^T$, with the determinant of its Jacobian being $\sin(\phi)$. Another example is solving 2D Poisson equation using Walk-on-sphere algorithm (Sawhney & Crane, 2020). In this case, we need to integrate over different largest 2D inscribed circles. In this case, we can define $\Omega(\mathbf{z})$ as $\{\mathbf{x} \in \mathbb{R}^2 | \|\mathbf{x} - \mathbf{z}\| \leq \text{dist}(\mathbf{z})\}$, where $\mathbf{z}$ is the center of the circle and dis returns the distance to the cloest point on the boundary. We can define the transformation $\Phi$ as $\Phi([r, \theta]^T; \mathbf{z}) = [\text{dist}(\mathbf{z})r\sin(\theta), \text{dist}(\mathbf{z})r\cos(\theta)]^T$, with $r \in [0, 1]$ and the determinant of Jacobian being $r \, \text{dist}(\mathbf{z})$.

For simplicity of notation, we will first discuss this problem by dropping the dependency of $\mathbf{z}$. We will then discuss how to incorporate $\mathbf{z}$ into the picture in Section 4.4. For a given domain $\Omega$ parameterized by $\Phi$, we can rewrite the integration into the following form by applying the change of variable formula:

$$F(\Omega) = \int_{\Omega} f(\mathbf{p}) d\mathbf{p} = \int_{l_1}^{u_1} \cdots \int_{l_d}^{u_d} f(\Phi(\mathbf{x})) |J_{\Phi}(\mathbf{x})| d\mathbf{x}, \tag{5}$$

122 where $J_\Phi$ denotes the Jacobian of function $\Phi$, which is a parameterization of the integration domain.

123

**Monte Carlo Integration** A common way to compute such integration numerically is via Monte Carlo methods (Veach, 1998). The main idea of Monte Carlo integration is to rewrite the integration into an expectation, which can be estimated via sampling. For example, to estimate Equation 5 with the Monte Carlo method, we first write it into an expectation over the domain $\Omega$ and estimate the expectation via sampling:

$$F(\Omega) = \int_{x \in \Omega} f(\mathbf{p}) d\mathbf{p} = \mathbb{E}_{\mathbf{p} \sim P_\Omega} \left[ \frac{f(\mathbf{p})}{P_\Omega(\mathbf{p})} \right] \approx \frac{1}{N} \sum_{i=1}^{N} \frac{f(\mathbf{p}_i)}{P_\Omega(\mathbf{p}_i)}, \quad \mathbf{p}_i \sim P_\Omega(\mathbf{p}), \tag{6}$$

124 where $P_\Omega$ is a distribution over domain $\Omega$ from which we can both sample points and evaluate
125 likelihood. While Monte Carlo estimation is unbiased, it usually suffers from high variance, which
126 requires a lot of samples and function evaluation of $f$ and $P_\Omega$ to reduce.

**Control Variates.** Control variates is a technique to reduce variance for Monte Carlo estimators. The key idea is to construct a new integrand with lower variance and apply Monte Carlo estimation for the low variance integrand only. Suppose we know $G = \int_\Omega g(\mathbf{p}) d\mathbf{p}$ for some $G$ and $g$, then we can create the following unbiased Monte Carlo estimator for the original integral of $f$:

$$F(\Omega) = \int_\Omega f(\mathbf{p}) d\mathbf{p} = c \cdot G + \int_\Omega f(\mathbf{p}) - c \cdot g(\mathbf{p}) d\mathbf{p} \approx c \cdot G + \frac{1}{N} \sum_{i=1}^{N} \frac{f(\mathbf{p}_i) - c \cdot g(\mathbf{p}_i)}{P_\Omega(\mathbf{p}_i)}, \tag{7}$$

127 where $\mathbf{p}_i$ are samples from distribution $P_\Omega$ and $c$ is any constant in $\mathbb{R}$. As long as $G$ is the analytical
128 integration result of $g$, the new estimator created after applying control variate is unbiased. Note
129 that the control variate estimator is running Monte Carlo integration on the new integrand $f(\mathbf{p}) -$
130 $c \cdot g(\mathbf{p})$, instead of the original integrand $f(\mathbf{p})$. The key to a successful control variate is finding
131 corresponding functions $G$ and $g$ that make $f(\mathbf{p}) - c \cdot g(\mathbf{p})$ to contain less variance compared to the
132 original integrand under the distribution $P_\Omega$. In this paper, we will demonstrate how to create a class
133 of $G$ and $g$ using neural integration techniques to achieve this goal.

**Neural Integration.** An alternative approach is to use a neural network to approximate the output of the integration, as introduced by AutoInt (Lindell et al., 2021). AutoInt trains a neural network $G_\theta(T)$ to approximate line integration of the form $\int_a^T f(x) dx$ for some fixed $a \in \mathbb{R}$. To achieve this, AutoInt leveraged the first fundamental theorem of calculus to derive the loss required to find the optimal $\theta^*$:

$$\theta^* = \arg\min_\theta \mathbb{E}_{x \in \mathcal{U}[L,U]} \left[ \|f(x) - G_\theta'(x)\|^2 \right], \tag{8}$$

134 where the derivative $G_\theta'(x)$ is obtained via the automatic differentiation framework and $L, U \in \mathbb{R}$
135 are two real-numbers defining the integration domain of interest. Once the network is trained,
136 we can use optimized parameters $\theta^*$ to approximate the integration results of $\int_l^u f(x) dx$ since
137 $\int_l^u f(x) dx \approx G_{\theta^*}(u) - G_{\theta^*}(l)$ for all $L \leq l \leq u \leq U$. This idea can be extended to multi-
138 variables integration (Maître & Santos-Mateos, 2023) by taking multiple derivatives, which we will
139 leverage in the following section to construct integration of a parameterized spatial domain.

140 Compared to Monte Carlo integration, neural integration can approximate a family of integral (i.e.
141 for all pairs of $(l, u)$ such that $L \leq l \leq u \leq U$) efficiently where each integration result can be
142 obtained with two neural network forward passes. However, it's difficult to provide guarantees
143 that the network $G_\theta$ can approximate the integration of interest accurately. It's generally hard to
144 ensure the loss reaches zero. In this paper, we propose to alleviate these issues by using the neural
145 technique as the Monte Carlo control variate, achieving unbiased and low-variance estimation.

## 4 METHOD

147 In this section, we will demonstrate how to combine Monte Carlo control variates technique with
148 neural integration techniques to estimate a family of spatial integrations. We will first demonstrate

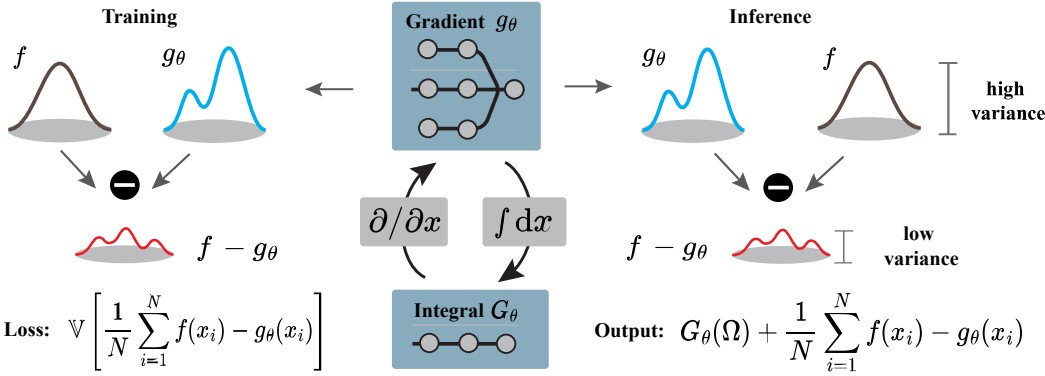

Figure 1: Illustration of method. We first create two computational graph $G_\theta$ and $g_\theta$ with shared parameter and property that $G_\theta = \int_\Omega g_\theta(\mathbf{p}) d\mathbf{p}$ (Sec 4.1, middle figure). During training, we optimize the parameter $\theta$ to minimize the variance of $f - g_\theta$ (Sec 4.3). During inference, we apply Monte Carlo estimation on $f - g_\theta$, which can have lower variance than the original integrand $f$ (Sec 4.2).

how to construct networks with known analytical spatial integrals (Sec 4.1) and how to create an unbiased estimator using these networks as control variates (Section 4.2). We will derive a loss function to minimize the variance of the neural estimator (Sec 4.3). Finally, we discuss how to extend this formulation to multiple domains (Sec 4.4) and how to choose architectures (Sec 4.5).

## 4.1 NEURAL AUTOMATIC SPATIAL INTEGRATION

In this section, we will show how to generalize the idea of neural automatic integration to multivariable spatial integration on a domain $\Omega$ parameterized by function $\Phi$. Let $\mathbf{u}_i$ and $\mathbf{l}_i$ represents the upper and lower bound of the integration for the $i^{\text{th}}$ dimension for all $i = 1, \ldots, d$. Let $G_\theta : \mathbb{R}^d \to \mathbb{R}$ be a neural network that approximates the anti-derivative of the integrand $f$. Now define the integral network $I_\theta : \mathbb{R}^d \times \mathbb{R}^d \to \mathbb{R}$ as the following:

$$I_\theta(\mathbf{u}, \mathbf{l}) = \sum_{(s_1, x_1) \in \{(-1, \mathbf{l}_1), (1, \mathbf{u}_1)\}} \cdots \sum_{(s_d, x_d) \in \{(-1, \mathbf{l}_d), (1, \mathbf{u}_d)\}} G_\theta(\mathbf{x}) \prod_{i=1}^d s_i, \tag{9}$$

where $\mathbf{x} = [x_1, \ldots, x_d]$. By the first fundamental theorem of Calculus, we have the following:

$$I_\theta(\mathbf{u}, \mathbf{l}) = \int_{\mathbf{l}_1}^{\mathbf{u}_1} \cdots \int_{\mathbf{l}_d}^{\mathbf{u}_d} \frac{\partial^d G_\theta(\mathbf{x})}{\partial \mathbf{x}_1 \ldots \partial \mathbf{x}_d} dx_d \ldots dx_1, \tag{10}$$

where $\frac{\partial^d G_\theta(\mathbf{x})}{\partial \mathbf{x}_1 \ldots \partial \mathbf{x}_d}$ is the $d^{\text{th}}$ order derivatives of $G_\theta$ computed using automatic differentiation for each dimension of $\mathbf{x}$. With slight abused of notation, we denote $\frac{\partial^d G_\theta}{\partial \mathbf{x}_1 \ldots \partial \mathbf{x}_d}$ as $\frac{\partial G_\theta}{\partial \mathbf{x}}$. Note that we can obtain both the computation graph for the integrand $\frac{\partial^d G_\theta}{\partial \mathbf{x}}$ and the approximation to the integral integral, $I_\theta$, using existing deep learning frameworks such as PyTorch (Paszke et al., 2019), Jax (Bradbury et al., 2018), and Tensorflow (Abadi et al., 2015). This allows us to leverage the AutoInt loss to learn parameters $\theta$ to approximate this integral using $I_\theta$.

This idea of automatic integration can be extended to handle integration over the domain $\Omega$ parameterized by a function $\Phi : \mathbb{R}^d \to \Omega$. To achieve this, we need to apply a change of variable to the previous equation using $\Phi$, mapping from the $[\mathbf{l}_1, \mathbf{u}_1] \times \cdots \times [\mathbf{l}_d, \mathbf{u}_d]$ space to $\Omega$:

$$I_\theta(\mathbf{u}, \mathbf{l}) = \int_{\mathbf{l}_1}^{\mathbf{u}_1} \cdots \int_{\mathbf{l}_d}^{\mathbf{u}_d} |J_\Phi(\mathbf{x})| \frac{\partial^d G_\theta(\mathbf{x})/\partial \mathbf{x}}{|J_\Phi(\mathbf{x})|} d\mathbf{x} = \int_\Omega \frac{\partial^d G_\theta(\Phi^{-1}(\mathbf{p}))}{\partial \mathbf{x}} |J_\Phi(\Phi^{-1}(\mathbf{p}))|^{-1} d\mathbf{p}. \tag{11}$$

Note that $\frac{\partial^d G_\theta(\Phi(\mathbf{p}))}{\partial \mathbf{x}} |J_\Phi(\Phi^{-1}(\mathbf{p}))|^{-1}$ is also a computational graph we can obtain through automatic differentiation from $G_\theta$. At this point, we are able to apply the idea of AutoInt to obtain $\theta$ that can make $I_\theta$ approximate integral $\int_\Omega f(\mathbf{p}) d\mathbf{p}$ by optimizing this following loss:

$$\mathcal{L}_{\text{autoint}}(\theta) = \mathbb{E}_{\mathbf{p} \sim P_\Omega} \left[ \left\| \frac{\partial^d G_\theta(\Phi^{-1}(\mathbf{p}))}{\partial \mathbf{x}} |J_\Phi(\Phi^{-1}(\mathbf{p}))|^{-1} - f(\mathbf{p}) \right\|^2 \right], \tag{12}$$

160 where $P_\Omega$ is a distribution over $\Omega$ that we can sample from. Once we obtained $\theta^*$ by running SGD
161 on $\mathcal{L}_{\text{autoint}}$, we can use $I_{\theta^*}$ to approximate the spatial integral.

## 4.2 Unbiased Estimation via Control Variate

Though we are now able to extend the AutoInt idea to spatial integration, the resulting network $I_{\theta^*}$ can still be biased. One way to achieve unbiased estimation is to use the neural network estimate as a control variate. Specifically, the integration can be written in the following form:

$$\int_\Omega f(\mathbf{p})d\mathbf{p} = I_\theta(\mathbf{u}, \mathbf{l}) + \int_\Omega f(\mathbf{p}) - \frac{\partial^d G_\theta(\Phi(\mathbf{p}))}{\partial \mathbf{x}} |J_\Phi(\Phi(\mathbf{p}))|^{-1} d\mathbf{p}. \tag{13}$$

Now we can create a Monte Carlo estimator $E_{N,\theta}$ to approximate the spatial integration:

$$E_{N,\theta} = I_\theta(\mathbf{u}, \mathbf{l}) + \frac{1}{N} \sum_{i=1}^{N} \left( f(\mathbf{p}_i) - \frac{\partial^d G_\theta(\mathbf{x}_i)}{\partial \mathbf{x}} |J_\Phi(\mathbf{x}_i)|^{-1} \right) P_\Omega(\mathbf{p}_i)^{-1}, \tag{14}$$

163 where $\mathbf{p}_i \sim P_\Omega$ are independent samples from a distribution on the domain $\Omega$, $P_\Omega(\mathbf{p}_i)$ is the prob-
164 ability density of point $\mathbf{p}_i$ according to distribution $P_\Omega$, $N$ is the number of samples used for the
165 Monte Carlo estimator, and $\mathbf{x}_i = \Phi^{-1}(\mathbf{p}_i)$.

166 While the estimator $E_{N,\theta}$ is unbiased, it can show higher variance than directly applying Monte
167 Carlo estimation to the original integrand $f$ if $\theta$ is not chosen intelligently. We will show in the next
168 section how to minimize the variance of such an estimator using deep learning tools.

## 4.3 Minimizing Variance

The variance of a single sample Monte Carlo estimator $E_{N,\theta}$ in Equation 14 can be computed as:

$$\mathbb{V}[E_{N,\theta}] = \frac{1}{N} \left( \left( I_\theta(\mathbf{u}, \mathbf{l}) - \int_\Omega f(\mathbf{p})d\mathbf{p} \right)^2 + \int_\Omega \frac{\left( f(\mathbf{p}) - \frac{\partial^d G_\theta(\mathbf{x})}{\partial \mathbf{x}} |J_\Phi(\mathbf{x})|^{-1} \right)^2}{P_\Omega(\mathbf{p})} d\mathbf{p} \right), \tag{15}$$

where $\mathbf{x} = \Phi^{-1}(\mathbf{p})$. Directly using this variance as a loss function is infeasible since we do not have analytical solutions for the term $\int_\Omega f(\mathbf{p})d\mathbf{p}$. Instead, it's feasible to obtain samples of $(\mathbf{p}_i, f(\mathbf{p}_i))$ where $\mathbf{p}_i \sim P_\Omega$. The idea is to use these samples to construct a good estimate for the network gradient $\nabla_\theta \mathbb{V}[E_{N,\theta}]$. To achieve this, we first rewrite $\nabla_\theta \mathbb{V}[E_{N,\theta}]$ as following:

$$\nabla_\theta \int_\Omega P_\Omega(\mathbf{p}) \frac{(I_\theta(\mathbf{u}, \mathbf{l}) - f(\mathbf{p})|\Omega|)^2}{|\Omega| P_\Omega(\mathbf{p})} d\mathbf{p} + \nabla_\theta \int_\Omega P_\Omega(\mathbf{p}) \left( \frac{f(\mathbf{p}) - \frac{\partial^d G_\theta(\mathbf{x})}{\partial \mathbf{x}} |J_\Phi(\mathbf{x})|^{-1}}{P_\Omega(\mathbf{p})} \right)^2 d\mathbf{p},$$

where $|\Omega|$ denotes the area or volume of the domain: $|\Omega| = \int_\Omega 1 \cdot d\mathbf{p}$. Given this expression, we can create a Monte Carlo estimator for the network gradient by optimizing the following loss function:

$$\mathcal{L}(\theta, \Omega) = \underbrace{\mathbb{E}_{P_\Omega} \left[ \frac{(I_\theta(\mathbf{u}, \mathbf{l}) - f(\mathbf{p})|\Omega|)^2}{|\Omega| P_\Omega(\mathbf{p})} \right]}_{\text{Integral loss} = \mathcal{L}_{\text{int}}(\theta, \Omega)} + \underbrace{\mathbb{E}_{P_\Omega} \left[ \left( \frac{f(\mathbf{p}) - \frac{\partial^d F_\theta(\mathbf{x})}{\partial \mathbf{x}} |J_g(\mathbf{x})|^{-1}}{P_\Omega(\mathbf{p})} \right)^2 \right]}_{\text{Derivative loss} = \mathcal{L}_{\text{diff}}(\theta, \Omega)}, \tag{16}$$

170 where the expectation is taken by sampling a minibatch of $\mathbf{p}$'s from $P_\Omega$, and $\mathbf{x} = \Phi^{-1}(\mathbf{p})$. We set
171 $P_\Omega$ to be the same distribution used in the existing Monte Carlo estimator. This allows us to use the
172 existing Monte Carlo estimator to generate training data. Specifically, for each Monte Carlo sample
173 step, we will record the tuple $(\mathbf{p}, P_\Omega(\mathbf{p}), f(\mathbf{p}), |\Omega|)$ to be used for the training.

## 4.4 Modeling a Family of Integrals

175 So far we've focused our discussion on modeling different outcomes of a single integration
176 $\int_\Omega f(\mathbf{p})d\mathbf{p}$ over a single domain $\Omega$. In many applications, we usually need to perform multiple
177 spatial integrals, each of which will be using a slightly different domain $\Omega$. Specifically, we are

interested in a family of domains $\Omega(\mathbf{z}) \subset \mathbb{R}^d$, where $\mathbf{z} \in \mathbb{R}^h$ is a latent variable that parameterizes these domains. We further assume there exists a family of parameterization functions for this family of domains $\Phi : \mathbb{R} \times \mathbb{R}^h \to \Omega$, where each function $\Phi(\cdot, \mathbf{z})$ is differentiable and invertible conditional on $\mathbf{z}$. We are interested in approximating the results for a class of integrals with integrand $f(\mathbf{p}, \mathbf{z})$: $F(\mathbf{z}) = \int_{\Omega(\mathbf{z})} f(\mathbf{p}, \mathbf{z})d\mathbf{p}$, for $\forall z \in \mathbb{R}^h$. To handle this, we will extend our network $G_\theta$ to take not only the integration variable $\mathbf{x}$ but also the conditioning latent vector $\mathbf{z}$. We will extend the loss function to optimize through different latent $\mathbf{z}$: $\mathcal{L}_{\text{multi}}(\theta) = \frac{1}{N} \sum_{i=1}^{N} \mathcal{L}(\theta, \Omega(\mathbf{z}_i))$.

## 4.5 ARCHITECTURE

Most network architectures are designed to be expressive when using forward computational graphs. Our method, however, requires a network architecture to be expressive in not only its forward computational graph but also when fitting its gradient to certain functions. This is because our loss function is composed of both the integral loss and the derivative loss (Equation 16). Our integral loss is trying to optimize a computational graph (i.e., $I_\theta$) containing a network forward pass toward an objective. The derivative loss is trying to shape a computational graph containing the derivative of $G_\theta$ (i.e. $\frac{\partial^d}{\partial \mathbf{x}} F(\mathbf{x})$)) to match an objective. This calls for an architecture with both an expressive forward computational graph and an expressive derivative computational graph. The latter requirement is usually overlooked in mainstream machine learning research. In this work, we found SIREN (Sitzmann et al., 2020) works best in practice for our applications. Specifically, for most of our experiment, we use concatenated SIREN in the following form:

$$G_\theta(\mathbf{x}, \mathbf{z}) = \mathbf{W}_n(\phi_{n-1} \circ \ldots \phi_0)([\mathbf{x}, \mathbf{z}]) + \mathbf{b}_n, \quad x_i \mapsto \phi_i(\mathbf{x}_i) = \sin(\mathbf{W}_i \mathbf{x}_i + \mathbf{b}_i), \quad (17)$$

where $\theta$ contains all $\mathbf{W}_i$'s, $\mathbf{b}_i$'s and $[\cdot, \cdot]$ concatenate two vectors.

## 5 RESULTS

In this section, we will provide a proof of concept for our method in scientific computing problems where spatial integration is needed. Specifically, we will apply our method to solve elliptic Partial Differential Equations. This has many applications in computer graphics, including image editing, surface reconstruction, and physics simulation. In this section, we'll demonstrate the result of our method in solving Laplace (Sec 5.2) and Poisson equations (Sec 5.1). We hope to show that our method is able to produce less variance than the naive Monte Carlo methods and achieve unbiased results, which is not achievable with existing neural network methods.

The baseline we're comparing with are Walk-on-Spheres solver and the AutoInt result from the trained network. In the context of solving PDEs, the Walk-on-sphere baselines can be thought of as directly applying Monte Carlo estimation to integrating $f(\mathbf{p})$. As for the AutoInt baseline, we will apply the same transformation as mentioned in Section 4.1 to obtain the integration network. Instead of using this integration network and its corresponding gradient network in the control variates way, the AutoInt baseline will directly output the result obtained by the integral network.

## 5.1 SOLVING 2D POISSON EQUATION

We apply our techniques to reduce variance on a Poisson equation over the domain $\Omega$:

$$\Delta u = f \text{ on } \Omega, \quad u = g \text{ on } \partial\Omega, \quad (18)$$

where the $\Omega$ denotes the 2D shape representing the domain we are solving the PDE over, $g$ is the boundary function, and $f$ is the forcing function. This equation can be solved by the integral form Sawhney & Crane (2020):

$$u(x) = \frac{1}{|\partial B_{d(x)}(x)|} \int_{\partial B_{d(x)}(x)} u(y)dy + \int_{B_{d(x)}(x)} f(y)G(x,y)dy, \quad (19)$$

where $d(x) = \min_{y \in \partial\Omega} \|x - y\|$ denotes the distance to the boundary and $B_r(c) = \{y \| \|y - c\| \le r\}$ is the ball centered at $c$ with radius $r$.

With this, Sawhney & Crane (2020) derives a Monte Carlo estimator for the Poisson equation:

$$\hat{u}(x_k) = \begin{cases} g(\bar{x}_k) & \text{if } d(x) < \epsilon \\ \hat{u}(x_{k+1}) - |B_{x_k}(x_k)|f(y_k)G(x_k, y_k) & \text{otherwise} \end{cases} \quad (20)$$

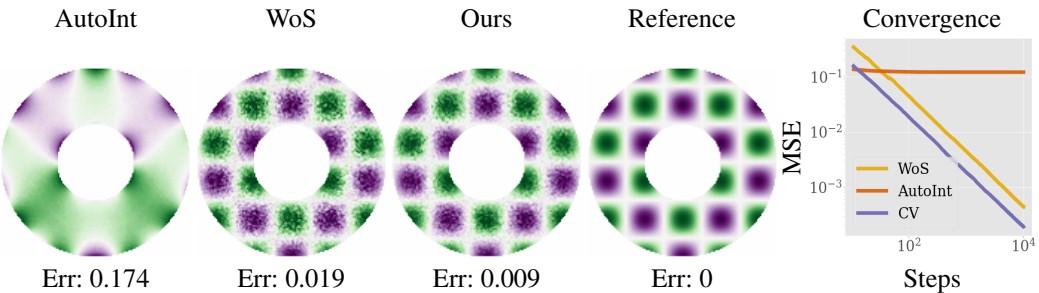

Figure 2: 2D Poisson solution on a Ring shape domain. Note that our method still produces lower variance than WoS even when the control variate integral network has bias.

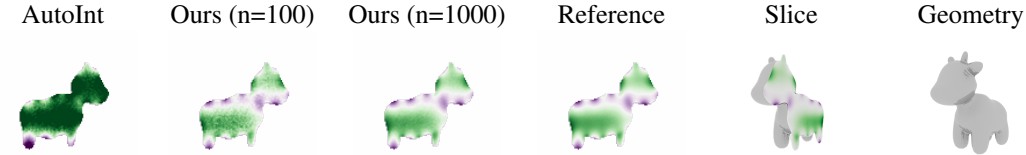

Figure 3: Result for 3D Laplace experiment. Both the AutoInt baseline and our method used the same network architecture and parameters. While the AutoInt baseline shows bias that is difficult to rectify with additional computes, our methods can create accurate solution when more compute is available to obtain samples, as suggested by the $n = 1000$ example being similar to the reference.

where $x_{k+1} \sim \mathcal{U}(\partial B_{d(x_k)}(x_k))$ and $y_k \sim \mathcal{U}(B_{d(x_k)}(x_k))$ are samples from the surface of the sphere and the inside of the sphere. These are two spatial integrals that our method can be applied to. For brevity, we are focusing on the sourcing part of the Poisson equation. However, our method can also be applied to the recursive part of estimating $u_y$, which will be investigated in detail in our next experiment that solves the 3D Laplace Equation

Applying our framework, we will train a SIREN network $G_\theta(s, x)$ with 128 hidden dimensions and 2 hidden layers, where $s \in \mathbb{R}^2$ is the polar coordinate and $x \in \mathbb{R}^2$ is the conditioning which modulates the integration domain: $\partial B_{d(x)}(x) = \{p \in \mathbb{R}^2 || p - x| = d(x)\}$, and $d$ is the distance function to the nearest boundary point. We'll train the network for $10^4$ iterations. At each step, we sample a one-step Monte Carlo estimator of the value $|B_{x_k}(x_k)| f(y_k) G(x_k, y_k)$ as our training label. We optimize it using our gradient network loss using the automatic differentiation framework Jax. Here's the estimator we used during the evaluation of the solver.

$$\hat{u}(x_k) = \begin{cases} g(\bar{x_k}) & \text{if } d(x_k) < \epsilon \\ \hat{u}(x_{k+1}) + |B_{d(x_k)}(x_k)| \left( f(y_k) G(x_k, y_k) - \frac{\partial G_{\theta^*}(x_{k+1})}{|J(x_{k+1})|} \right) + I_{\theta^*}(\vec{u}, \vec{l}; x_k) & \text{otherwise} \end{cases}$$

We present the qualitative result in an equal sample setting using a 2D ring geometry. As demonstrated by the qualitative images, our resulting image shows less noise than WoS solution and is more similar to the reference compared to the AutoInt solver. In addition, we also provide a convergence plot for this setting. Our method remains a $\log(1/N)$ convergence rate and preserves lower error than the WoS method when the AutoInt curve plateaus toward a biased value. This result verifies that our method can produce less biased results than the AutoInt baseline and also achieves lower variance than the WoS baseline.

## 5.2 SOLVING 3D LAPLACE EQUATION

In this section, we show that our proposed method can be used to reduce the variance of Walk-on-sphere (Sawhney & Crane, 2020; Muller, 1956) for solving Laplace equations:

$$\Delta u = 0 \text{ on } \Omega, \quad u = g \text{ on } \partial\Omega, \tag{21}$$

where $\Omega$ is the domain where we would like to solve the Equation equation. Sawhney & Crane (2020) shows that the solution of the Laplace equation can be expressed as the following integral equation: $u(x) = \frac{1}{|\partial B_{d(x)}(x)|} \int_{\partial B_{d(x)}(x)} u(y) dy$. Applying our framework, we will train a neural

network $G_\theta(s, x)$, where $s \in \mathbb{R}^2$ is the spherical coordinate and $x \in \mathbb{R}^3$ is the conditioning which modulates the integration domain: $\partial B_{d(x)}(x) = \{p \in \mathbb{R}^3 ||p - x| = d(x)\}$, and $d$ is the distance function to the nearest boundary point. Note that, different from the previous experiment, we're solving a recursive integration formula, so it's nontrivial to evaluate the integrand as it will spin up a series of random walks. At the same time, this is a series of spatial integrations, where we could apply our control variates on. We derive the following estimator:

$$\hat{u}(x_k) = \begin{cases} g(\bar{x}_k) & \text{if } d(x) < \epsilon \\ G_{\theta^*}(\vec{u}, \vec{l}; x_k) - 4\pi d(x)^2 \frac{\partial G_{\theta^*}(x_{k+1})}{|J(x_{k+1})|} + \hat{u}(x_{k+1}) & \text{otherwise} \end{cases} \quad (22)$$

where $x_{k+1}$ is sampled uniformly from the sphere centered at $x_k$ with radius $d(x_k)$, and $\bar{x}_k$ is the closest point of $x_k$ to the boundary. We will obtain $\theta^*$ by running Adam optimizer on loss in Equation 16. To obtain the data, we gather length-$k$ random walk sequence $x_0, \ldots, x_k$ that finally reaches the boundary with value $g(\bar{x}_k)$ using WoS solver. We use $g(\bar{x}_k)$ as a noisy (but unbiased) estimate for the training loss.

The result is presented in Figure 3. In this experiment, we use the same network parameter for our result and the AutoInt baseline. The left side of the figure shows that the result for the AutoInt baseline can be biased. Using the same network as AutoInt result, our method is able to create unbiased results when adding more computers to the inference time.

## 5.3 ABLATION

In this section, we conduct a series of ablation experiments within the context of solving a 2D Poisson Equation within a square domain. We mainly explore the (1) impact of different network architectures, specifically a concatenated version of SIREN and Random Fourier Features(RFF). (2) different sets of loss functions. In particular, we'll be looking at the loss that minimizes variance (Equation 16) and the AutoInt loss (Equation 12). Results of the ablations are shown in (Figure 4).

We observe that all of these trained control variates methods produce $\log(1/N)$ unbiased estimate. However, when using the same type of training loss, the SIREN network architecture shows a clear advantage over RFF, which was suggested by the Lindell et al. (2021) in the AutoInt but does not work well for our applications. In the meantime, the results show that minimizing variance as a training loss produces more accurate results.

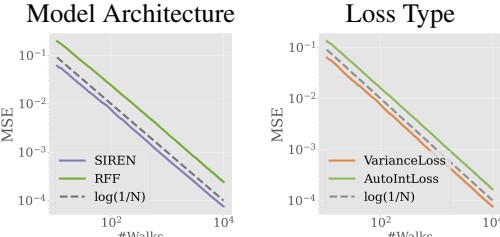

Figure 4: L: Ablation on model architecture using the same type of loss. R: Ablation on loss type using the same SIREN network architecture.

## 6 CONCLUSION

In this paper, we propose a method to approximate a family of spatial integration by combining neural integration techniques and Monte Carlo techniques. Our proposed method can potentially combine the merits of both methods - being unbiased as the Monte Carlo method while remaining low variance as the neural integration method. This is achieved by using the network produced by using the neural integration techniques as the control variate for a Monte Carlo sampler. To produce a low-variance estimator, we derive a loss function that can directly minimize the variance of the proposed estimator. We empirically test this idea on Monte Carlo PDE solvers and provide the proof of concept results showing that our proposed estimator is unbiased and can have lower variance compared to naive WoS estimators. Our method imposes very little restriction on architectural design. This can potentially open up an additional doorway that connects deep learning methods with Monte Carlo methods, inspiring innovation of new methods and applications.

**Limitations.** While the control variate Monte Carlo estimator is unbiased and potentially has low variance, such an estimator requires strictly more computation for each sampling step. This

Table 1: Time (in minutes) required to reach MSE to be less or equal to $3e - 4$.

| AutoInt | WoS | Ours |
|---------|-------|-------|
| 10.037 | 2.042 | 5.675 |

is because for every step, instead of evaluating $f$, we need to evaluate in additional $G$ and $g$ in order to produce the control variate estimator $G + (\sum_{i=1}^{N} f(x_i) - g(x_i))/N$. This suggests that the same improvement for the control variates obtained for the same amount of Monte Carlo samples might not translate to the performance improvement in actual compute, wall time, or energy, especially in simple settings (Table 1 provides some time profiling data). But we believe that in a more challenging integration setting, where the integrands $f$ is slow to evaluate or the probability distribution $P$ is difficult to sample, our proposed approach will be able to provide more advantages in wall time. Such mismatch in equal sample comparison is more severe when the compute taken to evaluate $g$ and $G$ is larger than the compute taken to evaluate $f$. This can limit the size of the network we can choose to express $G$. While applying automatic differentiation can construct analytical integration easily for various domains, it also requires taking multiple partial differentiations to create the network for training and inference. Taking the derivative of a network usually creates a larger computational graph, which adds to the issue of needing additional computing per sample. Computing the integration requires evaluating the network approximating the anti-derivative $2^d$ times, with $d$ being the dimension of the space we are integrating in. This limits our method's ability to scale to higher dimensions without additional care, such as Sun et al. (2023); Si et al. (2021) . Finally, while our loss provides a very good estimate of the gradient for minimizing the variance of the control variate estimator, the loss contains multiple division terms, such as division by the Jacobian. These can create numerical instability for training and inference.

**Future works.** Despite challenges, there are many opportunities in combining neural networks with Monte Carlo methods. One interesting direction is to leverage the flexibility to design new architectures curated to different applications and toward fixing different issues. For example, one can create a network architecture that is aware of the parameterization of the integration domain, which can leverage structures of the domain such as symmetry or other types of equivariances. Another interesting direction is to explore connections with other variance reduction techniques. For example, Müller et al. (2019) suggests leveraging importance sampling can propose training samples to allow efficient sampling. Other interesting directions include using these neural techniques as carriers to perform inverse graphics. Finally, it's interesting to extend this technique to other applications that require integration, such as image processing and rendering.

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
