# OpenReview forum: "AUTOMATIC NEURAL SPATIAL INTEGRATION"
_ICLR.cc/2024/Conference — Submitted to ICLR 2024_

### Official Review · Reviewer_PQTs · 2023-10-19

**Soundness:** 2 fair
**Presentation:** 3 good
**Contribution:** 2 fair
**Rating:** 5
**Confidence:** 3

**Summary:**

The paper constructs an estimator for the value of a multivariate integral.
More specifically, it constructs a multivariate version of ``AutoInt'' (Lindell et al.), a method of numerical integration using neural networks, and proposes to use it as a control variate in a Monte Carlo estimator.

**Strengths:**

In general, this is a decent paper.
I assess its biggest strength to be its readability:
The proposed ideas are reasonable and presented clearly. The exposition is easy to follow for anyone with prior exposure to the basics of Monte Carlo methods and neural networks (which should include almost everyone at ICLR).


However, I believe that the weaknesses (discussed next) do outweigh the strengths:

**Weaknesses:**

Even though this is a nice submission, I recommend rejection because I think that the contributions are, overall, too small.

For reference, I identify the following contributions:
- Deriving a multivariate version of ``AutoInt'' (Lindell et al.) via applying a multivariate fundamental theorem of calculus instead of a univariate one
- Using the resulting estimator as a control variate for Monte Carlo
- Training the ``AutoInt'' estimator by minimising the Monte Carlo variance instead of the loss used by Lindell et al.

The simulation results are okay, but the experiments lack a wall-time evaluation.
An improvement in error-per-steps compared to plain Monte Carlo and plain AutoInt alone is insufficient because the proposed estimator heavily relies on gradient information (whose computational costs are only unveiled by wall-time measurements).

My assessment would have been more positive if the contributions were more general or the simulation results were more convincing.
As is, I recommend rejection.

**Questions:**

1.  From reading the manuscript, it is unclear why using the estimator's variance as a target would be superior to the more common approach in the control-variate literature: constructing an approximation $g$, then estimating
$\int (f(x) -  c g(x)) dx + c \int g(x) dx$ while choosing $c$ such that the variance is minimal.
2. The curves in the "MSE per Steps" diagrams in Figures 2 and 4 look extremely straight/smooth. Is this an issue with the figure resolution, have they been smoothed/averaged before plotting, or is it simply a coincidence?
3. There is a typo at the bottom of page 6: " (...) applications. One interesting Specifically, for most (...)"

---

> ### Author Response · Authors · 2023-11-22
> **Response to Reviewer PQTs**
>
> We appreciate the comment on the readability and simulation results. In response to your inquiries, we would like to provide detailed explanations below:
> ### 1. General Contributions: (Please refer to General Response for more details)
> Thanks for recognizing the technical contributions of our paper. As for our main contribution, our goal is to introduce the idea of combining the Neural Field method with the Monte Carlo method using Control Variates. This contribution is very general as it can be applied to any spatial integration setting where the domain is structured. Unlike previous neural control variates methods where the networks are restricted to specific types of architectures, our methods impose minimal restrictions on the choice of network architectures. This allows users to choose more powerful and expressive network architectures, potentially leading to better numerical integration tools in the future. We believe that our contributions add knowledge to the field of numerical integration and can be beneficial to users with various applications in mind.
>
>
> ### 2. Wall-Time-Comparison:
> Since our current implementation is based on Jax, which is highly under-optimized for both WoS and Neural networks, wall-time comparison might not be very reliable. It also depends on the hardware where we run the experiments in, as well as what kind of PDE and neural networks we used for the experiment. As a result, we prioritized to present equal sample comparison, as done by many papers (Sawhney et al. 2022).
>
> Per the reviewer’s request, we’ve added the of our wall-time breakdown of our methods in comparison with the baselines, alongside our discussion of wall-time performance in the limitation sessions (Table 1 and L262-L265). Right now our proposed method takes more time compared to WoS to reach a certain MSE threshold in solving the 2D Poisson equation. We believe that the current profiling can be greatly improved if we investigate different ways to make neural network inference faster, such as implementing customized kernels. We also hypothesize that our method’s advantage will be more obvious when applying to solve more complicated PDEs, where WoS baseline will require more time per evaluation of integrands $f$ or per sampling from the domain.
>
>
> ### 3. Optimizing a network $g$ or just $c$ with a fixed, approximated $g$:
> If we only optimize $c$ with a fixed $g$, then the error between the shape of $f$  and the shape of g cannot be fixed. On the other hand, optimizing the network $g$ allows us to match f not only in scale but also in shape. As a result, optimizing a network $g$ can be more powerful to reach lower variance.
>
> Let’s examine the following example. Let $f(x) = x^2$ and $g(x) = x$. No matter how we optimize the $c$ we cannot reduce the variance of a Monte Carlo estimator for $\int_0^1 f(x) - cg(x) dx$ to zero. With our methods, we can define $G(x) = a_0 + a_1 x + a_2 x^2 + a_3 x^3$, where $a_0, \dots, a_3$ are optimizable parameters. We can see that $g(x) = G’(x) = a_1 + 2 a_2x + 3a_3 x^2$. One way to minimize the variance is to set $g(x) = f(x)$, namely $a_1 = 0, a_2=0, a_3 = 1/3$. This way our method can achieve zero variance, which is not possible if we only optimize with a fixed $g$ of choice: $g(x) = x$. In other words, our methods allows the data to pick the appropriate choice of $g$, which can be more efficient comparing to prior methods where $g$ is chosen via some (potentially wrong) heuristics.
>
> ### 4. Plotting issues:
> The actual result has a few slight bumps accumulating for 10k steps, you can zoom in to see the small bumps.

---

### Official Review · Reviewer_6qeM · 2023-10-29

**Soundness:** 3 good
**Presentation:** 3 good
**Contribution:** 2 fair
**Rating:** 3
**Confidence:** 4

**Summary:**

This paper proposes a method to approximate a family of spatial integration by combining neural integration and Monte Carlo techniques.

**Strengths:**

Using neural networks for integral estimation appears to be new in the literature, although it is a natural extension of the existing methods that use neural networks to estimate parameters for complex models.

**Weaknesses:**

1. It lacks theoretical guarantees for the performance of the proposed method. For example, the variance and bias of the estimator are hard to be assessed since the structure of the neural network used for estimation is undetermined.

2, The numerical experiments in the paper are limited.

**Questions:**

1. The author(s) mentioned that the variance of the Monte Carlo estimator decays at the rate of O(1/N) and thus a large number of independent samples are needed for accurate estimation. What is the rate for the proposed method?

2. Given the limited numerical experiments, it is hard to conclude that the proposed method can generally produce more accurate estimation than Monte Carlo. What are the main application scenarios of the proposed method?

---

> ### Author Response · Authors · 2023-11-22
> **Response to Reviewer 6qeM**
>
> Thank you for your thoughtful review! We would like to address some of your concerns as follows:
> ### 1. Convergence Rate of our method:
> The convergence rate of our proposed method is also O(1/N). This is shown in Figure 2 Convergence Plot and Section 5.3 (L233-L235)
> ### 2. Application Scenarios:
> We have discussed this in the second paragraph of the introduction section(L21- L27). Applications include solving PDEs and physics-based rendering, etc. These applications greatly impact various industries including movie, gaming, and mechanical engineering. Specifically, in this paper, we demonstrate our method’s ability to produce more accurate and efficient PDE solvers for Poisson and Laplace equations, which are among the most fundamental PDEs.
> ### 3. Limited Numerical Experiments: (Please refer to the general response for more details)
> Per the reviewer's requests, we have provided more numerical experiment results. The additional results are consistent with those we provided within the paper and these results also support our claims that the proposed neural control variates estimator is unbiased and also low-variance. The additional results further suggest that our method can generalize to different 2D/3D domains and boundary conditions.
> ### 4. Lack of theoretical guarantees of variance:
> We would like to clarify the misunderstandings of the variance guarantee. Our method inherits the merits of Monte Carlo methods - unbiased and with predictable variance. As shown in the paper shows that our method’s variance can be proved to decrease in O(1/N)  (Equation 15 and L233-L235)
>
> Furthermore,  our network is directly optimized to reduce the variance, which we’ve shown in Equation 16. These advantages come with very few restrictions on the network architectures and parameters except for letting the network be n-th time differentiable, which we believe greatly expands the choices of functions for control variates.

---

### Official Review · Reviewer_tiF6 · 2023-10-30

**Soundness:** 2 fair
**Presentation:** 1 poor
**Contribution:** 2 fair
**Rating:** 3
**Confidence:** 3

**Summary:**

This paper is devoted to the task of computing an integral $\int_{\Omega} f(p)dp$ over a known parameterized domain, where one has access to samples from a distribution over $\Omega$. The authors propose to combine monte-carlo integration methods with neural integration techniques; where neural integration techniques are used to reduce the variance of monte-carlo sampling.

More formally, the authors approximate the integral, by parameterizing a neural network $G_\theta:\mathbb{R}^d\to \mathbb{R}$. The authors assume the existence of a change of variable form a box constraint $\prod [l_i, u_i]$ to $\Omega$, $\Phi$. The authors extend auto-int to the multivariate case, by finding $\theta$ that such that $\frac{\partial^d G}{\partial x_1\dots\partial x_d}$ approximates well $f(p)$, by minimization of the MSE $\mathbb{E}\_p\|\frac{\partial^d G}{\partial x\_1\dots\partial x\_d}(\Phi(p))|D \Phi(p)|^{-1} - f(p)\|^2$.

The main idea of the authors is to then use this estimation of the antiderivative as a control variate to then do Monte-Carlo sampling rather than using it directly.
Indeed, assuming that the bounds $u$ are well-chosen, we have $\int f(p)dp  = G_\theta(u_1, \dots, u_d) + \int[ f(p) - \frac{\partial^d G}{\partial x\_1\dots\partial x\_d}(\Phi(p))|D \Phi(p)|^{-1} ]dp$. The authors then propose to approximate the last integral with monte-carlo sampling, and, therefore minimize as a loss its variance.

The authors then demonstrate that the proposed method works better than vanilla auto-int for solving 2-d poisson equations and 3-d laplace equation.

**Strengths:**

First, I want to mention that I have very little knowledge of numerical integration and Monte Carlo methods.

The main strength of the paper is that the proposed method seems sound and novel. It combines two well-known methods into a new one and demonstrates that it improves over both.

**Weaknesses:**

The principal weakness of this paper is that it is exceptionally unclear. There are many typos, the notations constantly change, some important concepts are not defined, some concepts are introduced twice, and some notations are confusing. This makes the full paper very hard to read. In this current state, this is clearly not fit for publication and requires a major rework before it becomes readable.

In my view, the amount of edits required to make the paper readable is too large to make it acceptable, even if the authors promise to fix all the errors.

In terms of methods, the authors do not really discuss the computational complexity of the method: what if the dimension d is large? For instance, the extension of auto-int requires computing a d-th derivative; the corresponding computational costs should be discussed.

Here is a list of typos + misc remarks.
-In eq.1 , f has two variables z, p . Then the authors write f(p) at the end of hte paragraph. This confusion is present many times in the paper.
- vectors are randomly written with either in bold font or with arrows on top: in eq.1 we have bold, in eq.2 we have arrows
- Top of page 2: G is a function of (a, b), then the authors write G'(x). There is also a constant change of notation between G_\theta and F_\theta.
- In the intro the authors write G as a function of p while later it will only act on the boxes [l_i, u_i] with variable x.
- "as takes"
- Page 3, the paragraph "Neural Network Integration Methods" contains many redundant citations
- Eq.4: yet another notation, this is redundant with eq1
- $\Phi$ should be one-to-one, not only invertible.
- "a region of R^d" -> the region should be a box constraint.
- $\Phi$ becomes $\Phi^{-1}$ later in the paper. The authors write $\Phi(x)$ in eq.5 but $\Phi(p)$ in e.g. eq11, 12
- "with Jacobian being r." It is the determinant of the Jacobian
- "once can see"
- around eq.8 : why all the notations "a, b , T, T_min, T_max"...
- below eq.8 $F'_\theta$.
- "(i.e. for all pairs of" missing closing parenthesis.
- The notion of anti derivative should be recalled precisely in the context  of multi-dimensional functions
- The notation $\frac{\partial^d G}{\partial x}$ should be explained clearly
- The authors write $\Phi(dp)$ in many places, dp should just be p.
- In the big eq. above eq. 16, $I_\theta$ should be $G_\theta$.
- $F_\theta$ in eq. 16.

**Questions:**

- The authors write "preliminary results show that our proposed method is unbiased": how is it not axiomatic? Can the authors discuss this more?

---

> ### Author Response · Authors · 2023-11-22
> **Response to Reviewer tiF6**
>
> Thank you for pointing out all the typos and unclear notations. We have fixed them in the main paper and marked the change with a cyan color. We hope our revision can improve the mathematical rigorousness and clarity of our method section. We would also like to address the following questions:
> ### 1. Computational complexity when d is large:
> Thank you for asking this question. In our original paper, we discussed the computational complexity for different d: “Computing the integration requires evaluating the network approximating the anti-derivative $2^d$ times, with $d$ being the dimension of the space we are integrating in.” ( L271-L274)
>
> (Please refer to the General Response for details.)
>
> We've discussed how our method scaled with $d$ naively in the limitation in the paper. Empirically, our method could also be applied to higher dimensional problems with efforts currently: Our PDEs’ path integration domain is infinite dimensions. We demonstrate the applicability of our method by breaking down the infinite-dimensional integration domain into smaller dimensions. In other words, even though our techniques could not naively be applied to higher dimensional problems directly, as long as the problem could be broken down into smaller dimensions, our method would work out.
> In the meantime, we believe extending our method to be naively applicable in higher dimensions is an interesting future work problem. We’ve added this to the discussion in our current paper. (L274)
>
> ### 2. Axiomatic Unbiased:
> Thank you for pointing out this language issue. Our method is unbiased by construction. The experiments' results further confirm this as our MSE decays at the rate of  O(1/N) where $N$ is the number of samples. We’ve corrected the language issue in the paper in L55-L56.

---

### Official Review · Reviewer_X3ae · 2023-11-06

**Soundness:** 3 good
**Presentation:** 3 good
**Contribution:** 3 good
**Rating:** 5
**Confidence:** 4

**Summary:**

This paper focuses on using a combination of neural networks and Monte Carlo to estimate spatial integrals, with the neural network estimate serving as a control variate for Monte Carlo estimation. The paper briefly reviews Monte Carlo integration and the control variate technique, and then introduces their proposed method. The key idea of the method is to use a fairly flexible analytically tractable "neural" integral (with a tractable integrand corresponding to it) to create a control variate for a Monte Carlo estimate of the spatial integral. The method is successfully tested on the Poisson and Laplace equations both of which require fairly low-dimensional spatial integration. The paper builds on the AutoInt technique and improves it in a substantive way, convincingly demonstrating that the proposed technique removes more bias.

**Strengths:**

The main idea of the paper is solid and I think points in the right direction. Previous works have taken the neural estimate as the "primary" (and only) one, whereas the authors suggest to improve a Monte Carlo estimate with a control variate that is backed out of the neural estimate. This is a direction well-worth pursuing, and represents a methodological novelty, and (at least in my view) provides a good context for development of similar techniques. The paper, for the most part, is clearly written.

**Weaknesses:**

For a novice, the exposition can be improved, especially in 4.1 and 4.2. It would be nice to add an algorithm to the paper. I also think more details on the experiments can be provided. To the non-PDE audience, the experiments on solving a 2D Poisson equation and 3D Laplace equation seem somewhat out of context, so it would be good to explain, even if briefly, why this is an important problem.

**Questions:**

I appreciate the authors' statement on the limitation of the proposed method. Have the authors considered how to make the method more scalable? There is some related literature on control variates that may be relevant, e.g., https://arxiv.org/abs/2006.07487, https://arxiv.org/abs/2303.04756.

Follow-up question to the above. Can other parametric functions be used instead of the neural network?

Can the method be applied to other machine learning problems, e.g., problems of Bayesian inference? Have the authors considered this?

How sensitive is the estimator to sample size? Have you tried to run the method with different values of N?

---

> ### Author Response · Authors · 2023-11-22
> **Response to Reviewer X3ae**
>
> We greatly appreciate the insightful comments and suggestions provided by the reviewer. In response to your questions, we provide detailed explanations below:
>
> ### 1. Explain the importance of solving PDEs:
> Thank you for this suggestion! We originally motivated our applications in the introduction section (L21-L27). Per reviewer’s request, we also include a motivation in the result section to improve the flow (L189-L192 in the paper).
> ### 2. Scalability: (Please refer to the General Response for more details.)
> Thank you for providing the relevant papers. As noted in the general response, our experiment is a demonstration that our method can be applied to solve integration problems in higher dimensions. We believe that similar principle might be applicable to other applications where high-dimensional integrations can be broken down into a set of lower-dimensional integrations. We believe that exploring how to apply the principle proposed by our method to different applications will be great future work research. We’ve also included discussion in our revision of the paper. (L271-274)
> ### 3. Using parametric function instead of NN:
> The parametric function can be used. Some specific parametric functions (such as a weighted sum of polynomial kernels) have been studied by prior works (Salaün et al. 2022) (L349- L350).
> In the meantime, our methods open a wider variety of network architectural choices for control variates. We achieve this by proposing a more general way to construct the control variates pairs (i.e. the G and g such that G = \int g), placing minimal restrictions on the network architecture. This allows us to use not only differentiable parametric functions but also other powerful universal approximators such as deep neural networks with nonlinearities.
> ### 4. Bayesian Inference:
> We agree that extending our methods to Bayesian inference can be a very interesting research direction. One of the key insights of this paper is that when we need to construct a pair of functions for control variates, namely $G$ and $g$ such that $G = \int g$, we can first define a function to approximate $G$ and take its derivative fo find $g$. We believe that key insights like this can be potentially extended to other problems that might benefit from control variates, including Bayesian inference.
> ### 5. Result aggregating different $n$:
> In the 3D Laplace experiment setting (Fig.3 in the paper and provided above), we have provided an example with different sample sizes for our method. As the sample size became larger, our method results were closer to ground truth. Our error is decreasing in the rate of O(1/N)

---

### Author Response · Authors · 2023-11-22
**General Response**

# General Response:

We appreciate the reviewers’ agreement with the novelty (Reviewer X3ae, Reviewer tiF6, Reviewer 6qeM) and readability (Reviewer PQTs, Reviewer X3ae, Reviewer 6qeM) of our proposed method.

## Summary:
To recap, our paper proposes a novel method to estimate spatial integrations accurately and efficiently by combining both the Monte Carlo method and the neural integration method. Our method not only inherits the merits of Monte Carlo methods, i.e. being unbiased and with predictable error, but also opens the gateway to leverage the power of neural networks when constructing the function to reduce the variance. Specifically, our method allows choosing a wide range of different neural network architectures as control variates. Achieving this is nontrivial, as we have proposed several concrete technical contributions (as mentioned by reviewer PQTs), including an extension of the existing neural integration method to more general classes of integrations, such as multivariate spatial integration, and propose a loss function that can minimize the variance of the final estimator directly. Empirical results in Monte Carlo PDE solvers also support our claims, showing that our method can successfully produce a lower-variance estimator.

Although we still face some challenges (as we discussed in the limitation section in L255- L277), we believe that our paper adds to the knowledge of numerical integration methods that can benefit various applications of interests to the ICLR community, including solving PDE and conduct physics rendering.

We’ve followed the suggestions and completed all the suggested revisions in the main paper (revisions are marked in cyan). Here we will first respond to questions shared by reviewers.

### Scalability in higher dimensions(Reviewer X3ae, Reviewer tiF6):
Several reviewers are curious about the applicability of our method to higher dimensions. We appreciate that Reviewer X3ae has pointed out two relevant papers one can use to improve the scalability of our method. In fact, our experiment on solving PDEs leveraging Walk-on-Spheres(Monte Carlo Method) itself is an example of applying our method to solve higher-dimensional integration problems. Solving a PDE using Monte Carlo methods can be viewed as computing integration over all possible paths, which lives in an infinite dimensional domain. However, we were able to break down the problem into integration on a family of integration in two-or-three dimensions to make the problem solvable using our method. With this said, we have demonstrated that our method can be applied to solve higher-dimensional problems provided that one can transform the high-dimensional problem into a collection of multiple low-dimensional ones. We believe this is an interesting future work direction to leverage this insight to other interesting applications such as Baysian inference. We’ve included this discussion in the revised version of the paper (L271-274).


### Additional results (Reviewer X3ae, Reviewer 6qeM, Reviewer PQTs):
Per reviewers' requests, we have run more experiments with different domain shapes. The numerical results show that our method can generalize to different 2D/3D shapes. Here’s the Mean Squared Error for equal sample comparison of different methods:

| Domain Shape              | AutoInt | WoS    | Ours    |
|---------------------------|---------|--------|---------|
| 2D Pos - Square (n =1000) | 0.00047 | 0.0012 | 0.00037 |
| 2D Pos - Coin (n = 1000)  | 0.174   | 0.0010 | 0.0009  |

| Domain Shape   | AutoInt | Ours(n=100) | Ours(n=1000) |
|----------------|---------|-------------|--------------|
| 3D Lap - Bunny | 0.9408  | 0.0038      | 0.0004       |
| 3D Lap - Cow   | 0.08578 | 0.00056     | 0.00006      |

These two tables show that our method can be generalized to different domain shapes. These results are consistent with both the empirical results in our paper that a) our method could produce more accurate results compared to the WoS baseline even when our AutoInt networks are biased.  b) With additional computing, our methods can create more accurate results as n gets larger(as suggested by the n = 1000 example being closer to the reference compared to n = 100). These results also support the claim of our paper.

---

### Meta-Review · Area_Chair_MaQr · 2023-12-09

**Metareview:**

The four reviewers all agree that this paper is below acceptance threshold for ICLR. In particular, they believe that the contributions were on the weak side and that the writing of the paper could be improved to make it more clear.

**Justification For Why Not Higher Score:**

The reviewers were unanimous in this opinion.

**Justification For Why Not Lower Score:**

N/A

---

### Decision · Program_Chairs · 2024-01-16

Reject